# Leveraging body dielectric polarization for ambient electromagnetic energy recovery via e-textile

Yuanlong Li [1,2], Weifeng Yang [1,2] ✉, Alexander V. Shokurov [1] & Carlo Menon [1] ✉

Ambient electromagnetic energy surrounds us in daily life and holds great potential for powering the future distributed wearable electronics. However, current electromagnetic energy harvesting methods suffer from reflection losses and parasitic effects, typically generating only nanowatt-level power, which is insufficient for flexible electronic systems. Here, we propose a body dielectric polarization-enabled textile for electromagnetic energy recovery. By harnessing the space charge/ dipole polarization in the body, the textile can directly convert electromagnetic energy from skin surface without additional power generation entities. Due to the body dielectric polarization mechanism, the textile can recover up to 10.2 mW/person (10.01 dBm) from a common laptop during daily typing, surpassing most state-of-the-art electromagnetic energy harvesters. We further demonstrate the textile by continuously powering a wireless heart rate belt for overnight heart rate and ECG monitoring without a battery. This body dielectric polarization energy recovery strategy provides inspiration for future distributed flexible energy.

Over the past decade, smart textiles have emerged as an electronic platform for integrating sensors[1,2], displays[3,4], and other devices[5,6], enabling flexible body area networks (BAN) for applications in personal healthcare[7], medical treatment[8], and general human-machine interaction[9], while retaining a convenient, compliant fabric form-factor familiar to every human. However, powering these BAN systems with stable and sustainable energy remains a key challenge. Flexible batteries can provide stable power output but are limited by recharging cycles and lifespan[10,11], notwithstanding the potential risks of thermal runaway, mechanical deformation and chemical leakage associated with some types of batteries[12]. Wireless energy transmission technologies, such as inductive coupling[2,13], achieve high efficiency (~65%) but suffer significant losses when the receivers/antennas are misaligned or positioned at a non-optimal distance. Environmental energy harvesting approaches, including flexible photovoltaics[14], triboelectrics[15,16], and thermoelectrics[17,18], are of high potential in theory but are inherently limited by the performance of transducer

materials and the effect of environmental conditions (e.g., light intensity in case of solar cells, type and speed of movement for tri-boelectric generators, temperature and humidity) (Supplementary Table 1 and Supplementary Note 1). Moreover, these flexible power generation solutions are typically fixed at a specific location on the garment where its power output is required, potentially requiring multiples of such generators in case of multiple devices[19,20], which restricts the flexibility and distributed nature of BAN systems.

As an integrated platform of flexible electronics, the human body could potentially serve as a carrier for distributing energy from a single power terminal to multiple BAN components. If the energy can be obtained directly from the body itself, the BAN systems would no longer be constrained by a distributed power source or complex wire connection. In order to do so, the body needs to act as the collector of some ambient energy, which then would be transduced by the body into an electric current and distributed to the wearable BAN. Luckily, the everyday ambient environment of the human body is rich in stable

[1]Biomedical and Mobile Health Technology Laboratory, Department of Health Sciences and Technology, ETH Zurich, Lengghalde 5, Zurich, Switzerland. [2]These authors contributed equally: Yuanlong Li, Weifeng Yang. ✉e-mail: weifeng.yang@hest.ethz.ch; carlo.menon@hest.ethz.ch

electromagnetic (EM) energy[21,22], offering a promising energy source for the wearable distributed BAN systems. Current research on ambient EM energy harvesting primarily focuses on high frequency bands (MHz, GHz[23]). High frequency harvesting technologies rely on specialized electronic components, such as high frequency diodes[24,25], precise impedance matching[26], and optimized antennas[27,28] to minimize the signal loss and parasitic effects. Furthermore, the existing high-frequency EM energy harvesting technology provides quite low power levels (~nW level)[24,25] and thus can only be used for some ultra-low-power devices like LCD screens[24], making them inadequate for the energy demand of an entire BAN system. Apart from the high-frequency EM waves, our environment is also abundant in low-frequency EM fields (50/60 Hz) generated by power-line noise from household appliances and transmission lines, which also provides a potential energy source for the wearable BAN. These low-frequency signals are characterized by wavelengths (kilometer level) much longer than their transmission paths, which leads to reduced parasitic capacitance and radiation losses[29], and can be effectively captured through near-field induction[30–32]. In this regard, the human body, having a high average dielectric constant and conductivity ($\varepsilon \approx 10^6$ at 50 Hz; $\sigma \approx 0.1\,\mathrm{S\,m^{-1}}$)[33,34] as compared to air ($\varepsilon \approx 1$ at 50 Hz; $\sigma \approx 10^{-14}\,\mathrm{S\,m^{-1}}$), is an excellent candidate to be a transducer for EM energy transfer.

Here, we propose a body dielectric polarization-enabled textile (PolaTex) to recover ambient dissipated EM energy. This technology leverages the fact that the human body is a Gaussian surface, capitalizing on the spatial charge polarization of ions, as well as the dipole polarization of water molecules within the body, when the human wearing the PolaTex is exposed to low-frequency EM fields. Due to the body dielectric polarization mechanism, the PolaTex can recover up to 10.2 mW/person (10.01 dBm) from a common laptop during daily typing, surpassing most state-of-the-art electromagnetic energy harvesters. Furthermore, we demonstrate the practical application by powering a Bluetooth-based BAN system, enabling wireless, real-time heart rate monitoring. Our strategy provides a feasible route for future wearable flexible energy sources.

## Results

### Concept of body dielectric polarization-enabled body area networks

Textile-based body area networks (BAN) require multiple sensors and wireless transmission modules embedded in clothing or on the body to enable real-time monitoring and interaction[35]. However, this setup necessitates separate power supply nodes or complex distributed electrode designs, which conflict with the vision of smart clothing: wireless, battery-free, comfortable, and washable. In the everyday environment, power transmission lines and high-power electrical appliances (e.g., personal computers, air conditioners) generate stray EM fields, inadvertently radiating significant EM energy into the surroundings (Fig. 1a), which in turn can be a readily available energy source for the BAN. These leaked EM signals, resembling power-frequency noise, are primarily low-frequency (50/60 Hz), correspondingly long wavelength, and exhibit weak parasitic coupling[30]. Therefore, during the propagation of these low-frequency signals, both the energy loss caused by parasitic capacitance and the signal reflection effect will be significantly reduced.

Here, we propose a textile electronic device utilizing the body dielectric polarization mechanism for ambient EM energy recovery (Fig. 1a, b). This electronic textile consists of three layers: a Leica cotton textile substrate, a conductive coating and a low impedance hydrogel layer. By placing the electronic textile on the surface of human skin, the EM energy can be automatically recovered from the environment through the human body and transmitted to the textile. The EM energy recovery process includes three steps: (1) the ambient freely dissipated EM field creates a capacitive effect between the air and the human body, enabling charge accumulation; (2) the electric field partially polarizes ions and water molecules in the human body; (3) the textile induces a polarized electric field, recovering the energy from these charging effects to power the BAN. Through this polarization mechanism, the dissipated EM energy is continuously recovered by the human body and transmitted to a useful load via conductive textile on the body surface. As a result, electric energy can be directly obtained from the human body surface at the necessary anatomical site corresponding to a required BAN node position.

COMSOL simulations (Fig. 1c, d, Supplementary Figs. 1, 2 and Supplementary Note 2) compare the electric field distribution in air and near the human body. Due to the relatively high dielectric constant of the human body ($\varepsilon \approx 10^6$ at 50 Hz)[33], and conductivity ($\sigma \approx 0.1\,\mathrm{S\,m^{-1}}$), ambient EM field energy is concentrated at the body rather than dissipating in air, creating a measurable surface potential. For a 100 V transmitter, the voltage attenuates significantly within the 0–30 cm of air and continues to decrease with further distance. However, simulations show that if a human body is placed 30 cm from the transmitter, the decrease of voltage with distance slows significantly as it reaches the body surface (Fig. 1e). At 100 cm from the transmitter, the body still retains a surface potential of approximately 1 V, while the potential in the air, in the case where no human is present, approaches zero. As shown in Fig. 1f, in a typical office environment, we measured that the PolaTex can recover up to 10.01 dBm of ambient EM energy (50 Hz), while a suspended pair of electrodes (without body dielectric polarization effect) can only recover −30.63 dBm of energy (Supplementary Fig. 3). This effect can be harnessed to power small electronic devices, provided a well-conducting human-body compliant textile is used to transduce the energy harvested. For example, in this work, we demonstrate that the human body could directly power a wearable electronic watch in real-time without relying on batteries, just by recovering the EM energy emitted into the environment by an ordinary laptop (Fig. 2g). Details on the construction and operational principles of the PolaTex are provided further in the article.

### Principle of body dielectric polarization-enabled EM energy recovery

Why the human body allows for the recovery of the dissipated EM energy through polarization can be explained by Gauss's law in dielectrics. As illustrated in Fig. 2a, when radiated EM energy encounters the human body, it is equivalent to an electric displacement vector passing through the local closed Gaussian surface formed by the human body (body Gaussian surface). Then, the carrier inside the human body (containing water molecules as well as ions) will be polarized in the electrical field. Neglecting the effects of skin interface impedance, the bound charge (polarization charge) induced on the body surface can be expressed as:

$$Q_i = \oint (\varepsilon_r - 1) \cdot \varepsilon_0 \cdot E \cdot dA$$

where $\varepsilon_r$ is the dielectric constant of the human body; $\varepsilon_0$ is the vacuum dielectric constant; $E$ is the intensity of the electric field part of the EM field; $A$ is the surface area of the body exposed to the EM field. Dry air, the relative dielectric constant of which is approximately 1, cannot sustain polarization, resulting in an induced charge density of zero. In contrast, the human body, with a relative dielectric constant up to $10^6$ at 50 Hz[33], can convert a significant portion of EM energy as polarization energy, which can then be transduced into usable current by the electrode of the current collector. This polarization occurs in two primary forms: (1) space charge polarization of ions[36] and (2) dipole polarization of water molecules[37,38] (Fig. 2a).

In order to investigate the effect of the medium polarity on the ambient EM energy recovery, we first measured the absolute permittivity of various materials in the low-frequency range (4 Hz–8 MHz). Among these were dry air, pentane (a non-polar liquid), ultrapure water

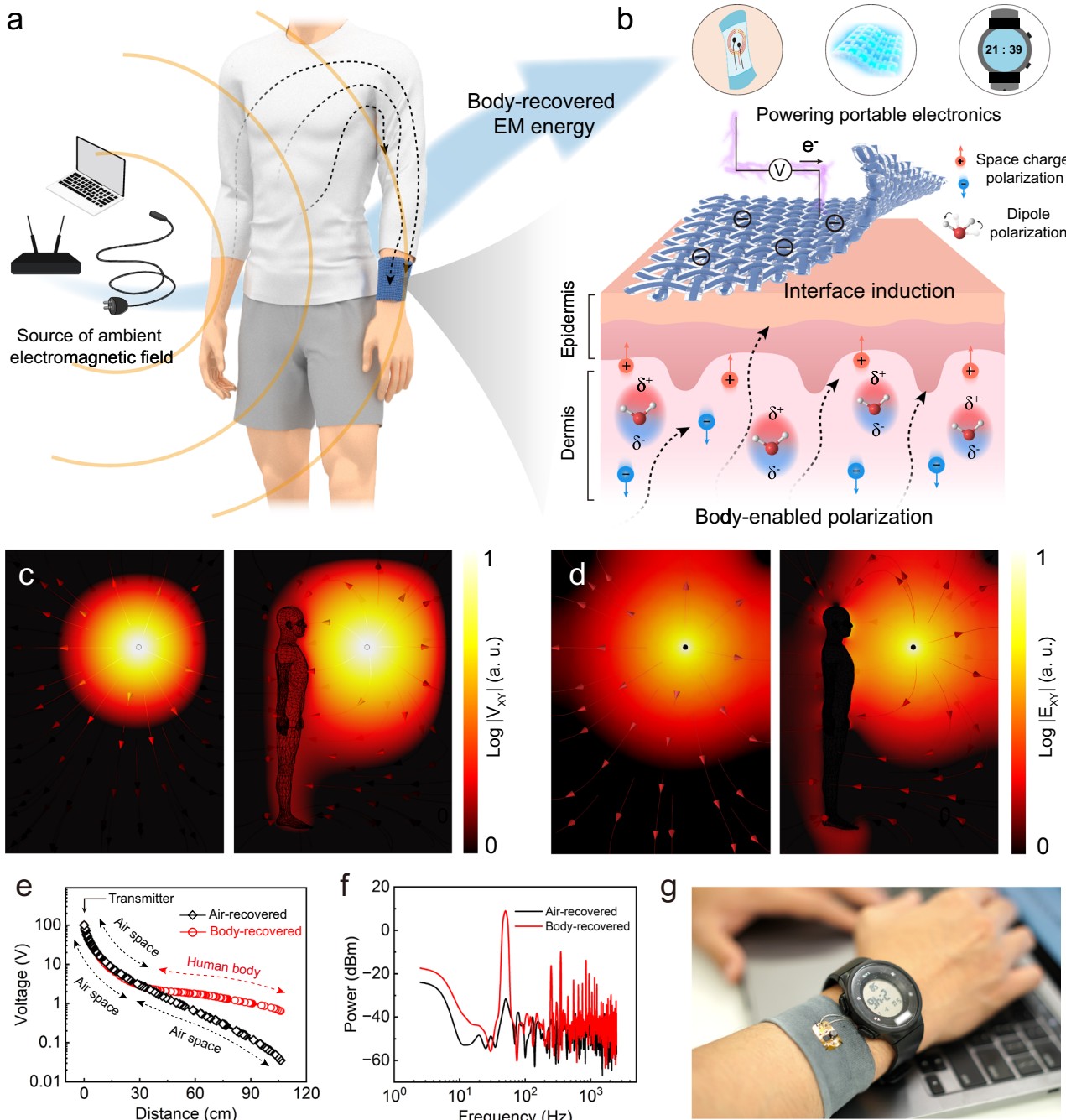

**Fig. 1 | Concept of body dielectric polarization-enabled body area networks.**
**a** Low-frequency EM energy dissipated by electronic equipment into the surrounding environment, including the human body, is a readily available source of energy to power wearable devices that are part of BAN. **b** Body dielectric polarization-enabled EM energy recovery to power the BAN. The illustration shows schematically how the ambient EM fields induce charge on the human body surface and how these effects can be used to generate electric current by PolaTex. **c** COMSOL simulation of voltage potential for air and human body. **d** A COMSOL simulation of the electric field for air and the human body. **e** COMSOL simulated potential curves showing the dissipation of energy of a 100 V transmitter in air, and at 30 cm from distance a human body. **f** A power spectrum of recovered EM energy under air and human body dielectric polarization conditions. The signal is measured directly from the daily typing office process without additional EM emission sources. Similar results were observed in three measurements. **g** Digital photos of the PolaTex powering an electronic watch through wireless power generation by recovery of energy emitted by the laptop.

(a polar liquid), and 1 M NaCl aqueous solution (Fig. 2b). Dry air and non-polar liquids exhibited minimal polarization and had the lowest absolute permittivity. In contrast, ultrapure water and ionic solutions displayed significantly higher absolute permittivity due to dipole polarization and space charge polarization. The high ion concentration in the ionic solution further enhanced space charge polarization. However, this effect diminished at higher frequencies due to ionic relaxation, as rapid electric field oscillations restricted ion migration.

As shown in Fig. 2c, d, dipole polarization is primarily determined by the total dipole moment of the dielectric molecules. For instance, water molecules (Fig. 2c) have an asymmetric structure, leading to a non-zero net dipole moment. When exposed to an electric field, these molecules align with the field. Non-polar molecules such as pentane (Fig. 2d), which have a symmetrical structure and no intrinsic dipole moment, do not react much with low-power low-frequency electric fields. To evaluate how molecular polarity affects EM energy recovery,

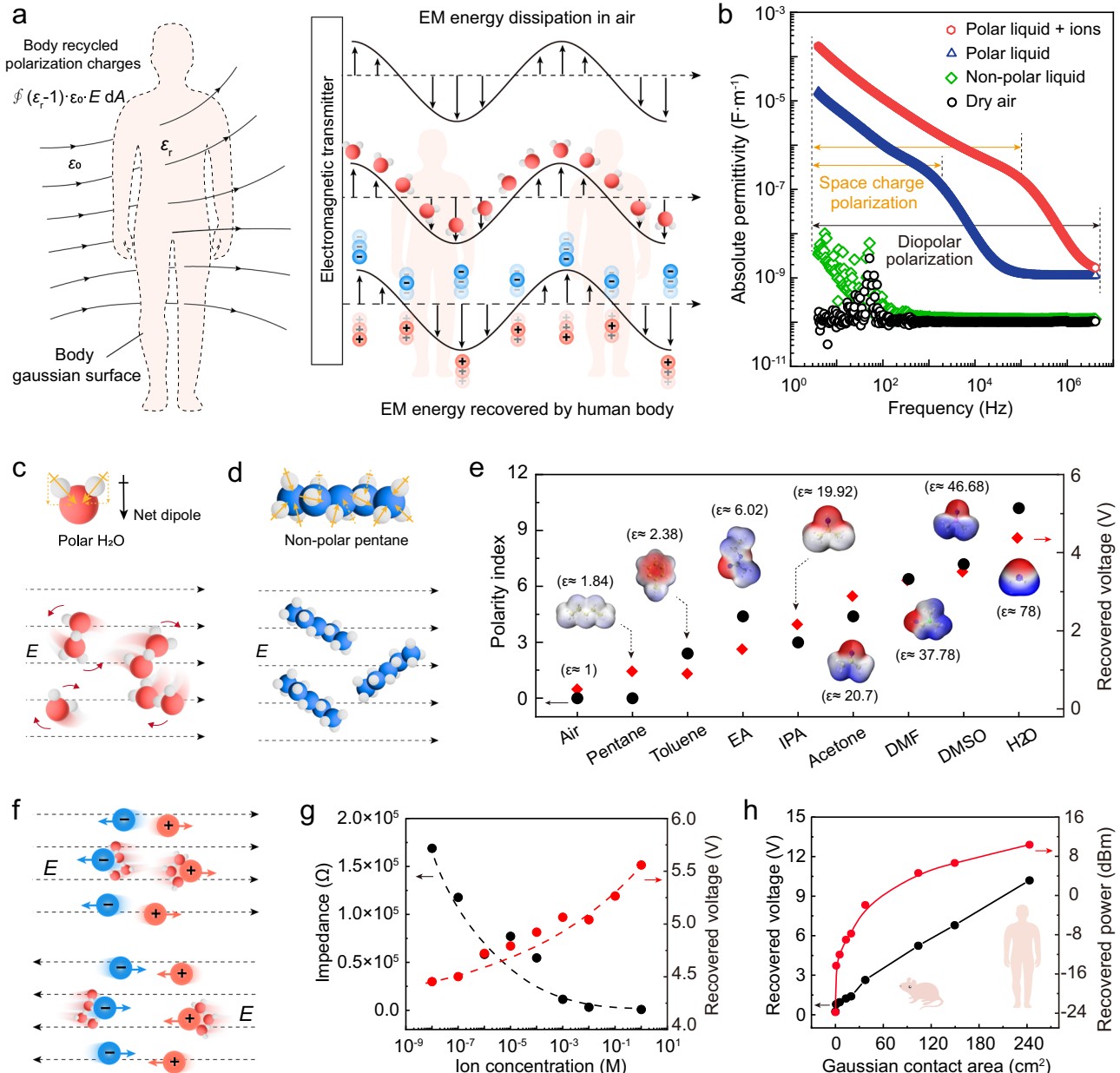

**Fig. 2 | Principle of body dielectric polarization-enabled EM energy recovery.** **a** A schematic diagram illustrating the interaction of the human body with the electric field, leading to polarized charges over the Gaussian surface of the human body. **b** Space charge polarization and dipole polarization lead to changes in the absolute permittivity of different dielectric materials at different frequencies. **c** Polar water molecules are deflected by an electric field. The same results were obtained in two independent tests. **d** Non-polar pentane molecules do not respond to electric fields. **e** The effect of the polarity index of the media on EM energy recovery. **f** Migration of charged ions under the action of an electric field results in space charge polarization. **g** Potential of aqueous solutions with different NaCl concentrations on EM energy recovery. **h** Dependence of recovered energy voltage and power as a function of exposed Gaussian surface area in the EM environment. Changing the bottom area of the reagent bottle to adjust the EM exposure area.

we compared solvents with different polarity index (Fig. 2e). We placed a reagent bottle containing a solvent on the laptop and inserted the oscilloscope probe into the solution to test the voltage signal (Supplementary Fig. 4 and Supplementary Movie 1). The results showed a positive correlation between a solvent's polarity index, dielectric constant, and the recovery voltage (Supplementary Fig. 5). Additionally, introducing NaCl at varying concentrations into ultrapure water (Fig. 2f and Supplementary Fig. 6) further increased the recovered voltage. Moreover, as the EM exposure area expanded, both the voltage and power of the recovered energy increased. Notably, even at small areas exposed to the EM fields (ca. 120 cm², which roughly corresponds to the body area of a small mammal), quite a lot of energy,

around 5 dBm, is being recovered, and even more for an exposure area comparable to the human body (Fig. 2h and Supplementary Fig. 7).

## Mechanical and electrical performance of body dielectric polarization-enabled textiles

Balancing electrical performance and wearable comfortability is crucial for flexible energy textiles. In order to reduce the interfacial impedance of the textile electrode while maintaining its good air permeability, we used air spraying to controllably deposit water-retaining hydrogel on the surface of PEDOT: PSS textile (Supplementary Fig. 8 and Supplementary Movie 2). First, PEDOT: PSS was impregnated onto the cotton surface, imparting electronic

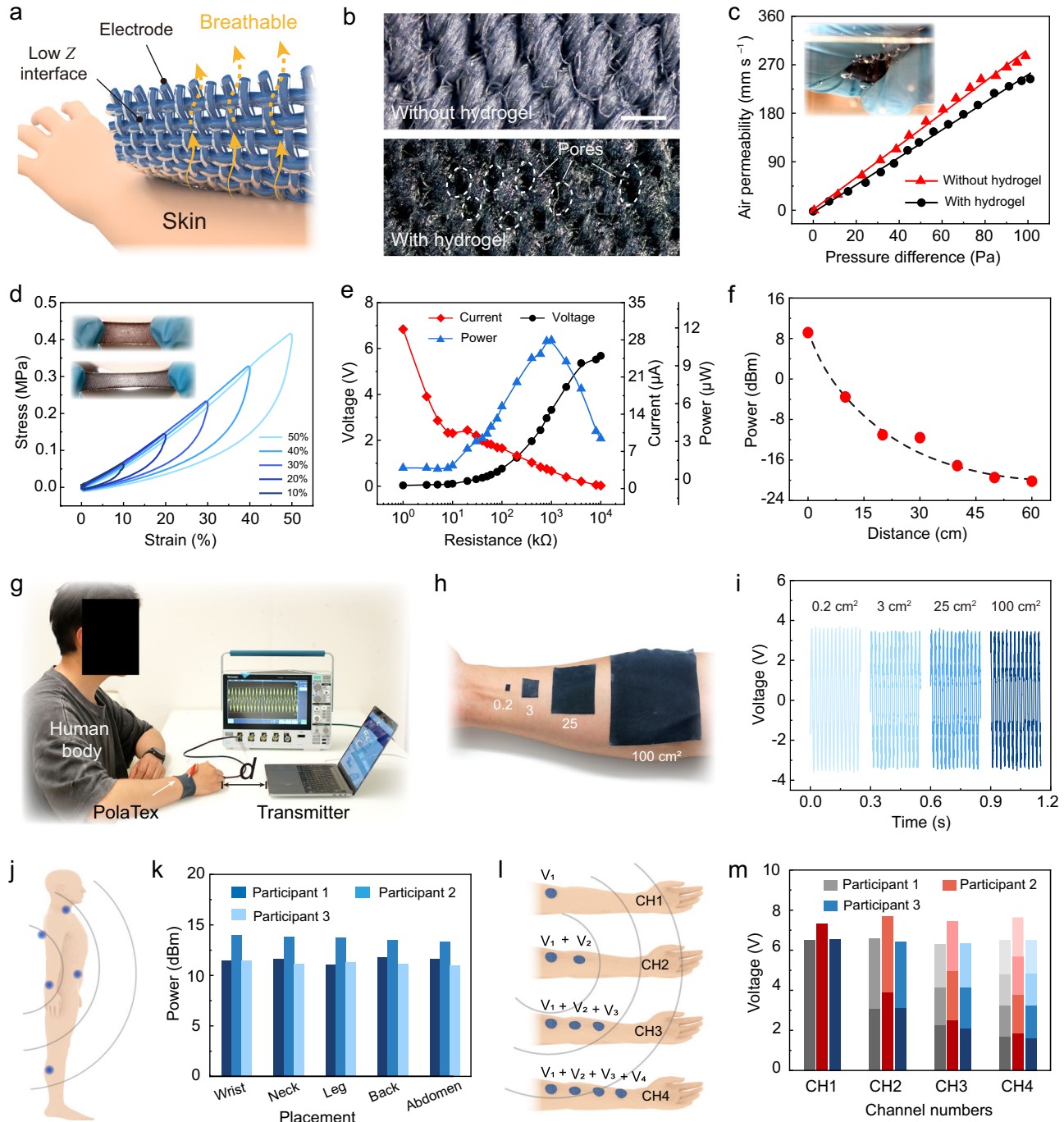

**Fig. 3 | Mechanical and electrical performance of body dielectric polarization-enabled textile. a** Schematic diagram of PolaTex on the skin. **b** Microscopic images of the cotton electrode before and after hydrogel treatment, scale bar represents 0.5 mm. The images provided here are representative of micrographs obtained from three samples. **c** Comparison of air permeability between hydrogel-treated and untreated textiles. **d** Stress-strain curves obtained by sequential multistep stretching and relaxing of hydrogel-modified electronic textiles. The stretching was performed at a fast rate of 5 mm/s and zero downtime between cycles, showing the mechanical properties of the elastic textile. **e** Electrical output performance of EM recovery under different load resistances. Power density was tested at 50 cm from the laptop. **f** Comparison of recovered EM energy at different distances. **g** Digital photo of PolaTex testing the electrical signal for recovering EM energy. **h** Digital photos of different areas of PolaTex (0.2 cm², 3 cm², 25 cm², 100 cm²) placed on the skin surface. **i** The voltage of EM energy recovered by PolaTex of different areas under the same conditions. Similar results were obtained for PolaTex samples in three independent tests. **j** Schematic diagram of EM energy recovery at different node positions on the human body. **k** Comparison of EM energy recovery at different nodes among different participants. The signal was measured during daily office typing while touching the laptop. **l** Schematic diagram of EM energy recovery with different numbers of nodes on the human body. **m** Comparison of recovered EM energy at different node numbers on the human body of different participants, segments of the columns represent the output of each individual node.

conductivity to the textile (Supplementary Figs. 9, 10). Second, a gelatin-based hydrogel layer was deposited on the PEDOT: PSS-treated cotton via air-spraying to reduce the skin-electrode interface impedance (Fig. 3a and Supplementary Figs. 10–13). By controlling the air-

spraying and application parameters, we could make sure that some pores of the textile remain intact in the uncovered regions, making sure that the Lycra cotton maintains its breathability (Fig. 3b and Supplementary Fig. 11c). Then, we soaked the hydrogel electrodes in

ammonium sulfate solution to further improve the mechanical properties and conductivity of the hydrogel interface based ion exchange[39]. In the air-permeability test (Fig. 3c), it was observed that the breathability of the textile remains largely unchanged before (~286 mm s⁻¹) and after (~243 mm s⁻¹) the hydrogel addition. We characterized the washability of PolaTex. After 10 washing cycles, the hydrogel layer on its surface remained well adhered to the textile substrate, and the electrical signal exhibited no significant attenuation (Supplementary Fig. 14). We also added different proportions of glycerol to improve its water retention, so that it can work stably within 2 - 3 days (Supplementary Fig. 11d). After PolaTex loses water, it can be restored to its initial state by soaking it in an aqueous solution for half an hour (Supplementary Figs. 15, 16). The PolaTex also showed stable elastic behavior in the stretching tests from the original length to 50% strain (Fig. 3d and Supplementary Figs. 17, 18). In addition, PolaTex exhibits excellent moisture and thermal stability, maintaining stable environmental EM energy recovering performance under conditions of sweating (Supplementary Fig. 19), high temperatures (Supplementary Fig. 20a–c) and high humidity (Supplementary Fig. 20d, e).

Power density is a key metric for EM energy harvesting devices. As shown in Fig. 3e, an output power of 11.02 μW/person was achieved under a load resistance of approximately 1 MΩ. The power density test process is as follows: a person wearing the PolaTex, sitting at 50 cm from the laptop, and then the voltage signal is measured from the fabric electrodes by an oscilloscope under different external loads (See "Methods"). When we directly touched the laptop for daily typing, the power density obtained is as shown in Supplementary Figs. 21, reaching up to 6.05 mW/person (Calculated based on $U^2/R$, where U is voltage, R is load resistance). The recovered energy gradually decreased as the distance from the electrical device increases (Fig. 3f, g and Supplementary Figs. 22, 23). Increasing the size of PolaTex has no obvious effect on the recovered voltage (Fig. 3h, i), while the exposed area of the recycled object has a significant effect on the voltage (Fig. 2h). Due to individual physiological differences, the amount of EM energy recovered varied from person to person, even under identical environmental and positional conditions (Supplementary Figs. 24). However, for a particular individual, the recovered energy remained relatively consistent, regardless of the electrode placement, with only minor variations. The testing locations included the wrist, leg, and abdomen (Fig. 3j and Supplementary Figs. 25), which suggested that wearable electronic devices placed on the wrist or abdomen can efficiently utilize the recovered energy (Fig. 3k and Supplementary Figs. 25). Moreover, when multiple nodes operated simultaneously (Fig. 3l), the energy recovered at each node remains comparable to that of an independently operating single node (Fig. 3m). This enabled simultaneous energy harvesting across multiple nodes without mutual dependence, ensuring that the energy distribution among different nodes does not compromise overall device performance. We discussed the biological safety of PolaTex under 50 Hz EM field. Both the potential Joule heating effects and electromagnetic radiation exposure were assessed, and no significant hazards were observed under current experimental conditions (Supplementary Tab. 2 and Supplementary Note 3).

### Application of body dielectric polarization-enabled textiles in body area networks

Current energy supply solutions typically deliver external energy to energy-consuming terminals through physical energy conversion units, such as coils and power generation devices. However, BAN has many nodes requiring many energy supply entities at each node, which inevitably brings the challenge for seamless integration of distributed energy. The technology proposed in this work tackles this challenge by allowing energy recovery and distribution via the BAN-carrying object (human body) itself.

The energy flow of our body dielectric polarization-enabled technology can be summarized as follows: (1) ambient EM energy is recovered by the human body; (2) the body itself is utilized as a pathway for energy transmission; (3) energy obtained via the PolaTex is then delivered to the corresponding energy consumption position of the BAN, e.g., a wearable device. (Fig. 4a, b). Since the placement of energy-consuming devices is inherently determined by the human body, this method of harnessing energy from the body eliminates location constraints and effectively addresses the distributed energy challenge in body area networks (Fig. 4c).

In practical applications, we demonstrate the PolaTex for powering the heart rate monitoring (HRM) device (Fig. 4d and Supplementary Movie 3). By using PolaTex, the ambient EM energy was recovered and rectified into direct current (DC) by an energy management module, which then charged a 10-mF capacitor. This capacitor can power the HRM when its voltage reaches 3 V. The HRM monitors the electrocardiographic (ECG) signals to calculate the heart rate (HR) and transmits the data via Bluetooth to a smartwatch for real-time monitoring (Supplementary Fig. 26). The HRM device, as shown in Fig. 4f, did not contain an external power supply in its circuit and relied solely on body dielectric polarization-enabled EM energy recovery as its power source. Using this approach, we have monitored HR and respiratory rate data overnight (Fig. 4g), where a significant increase in heart rate was observed after waking up. Moreover, the body dielectric polarization-enabled EM energy recovery technology can be applied in a multi-node configuration at different body locations if needed (Fig. 4e). It can be integrated into PolaTex at various positions, such as foot, knee, and wrist nodes as socks, knee-sleeves, and wristbands, respectively.

In addition, we also developed a flexible energy management circuit to stabilize the output voltage at 3.3 V to match the rated voltage of some commercial electronic equipment (Fig. 4g–j and Supplementary Fig. 27). The equivalent circuit of energy management is shown in Fig. 4k. Its main functions are rectification, load protection, energy storage and voltage regulation output. We also compared the effect of diodes with different reverse leakage currents on the final recovered EM energy (Supplementary Fig.28 and Supplementary Tab. 3), which is then rectified to charge a 47 μF capacitor. By connecting the wristband to a light-emitting fiber device (Fig. 4l, m and Supplementary Movie 4), it can continuously power the light-emitting fibers during use, enabling operation without an external power source. Furthermore, the wristband-based power supply device can be used to charge an electronic watch (Supplementary Fig. 29 and Supplementary Movie 5). We used an oscilloscope to record the voltage-time curve of the complete process of recovering electromagnetic energy and driving electronic devices (Fig. 4n). When powering different electronic devices, the final output voltage varies depending on the power consumption of each device. In a control experiment where the same device was not in contact with the human body, the energy recovered approaches zero, making it essentially incapable of powering electronic devices, and thus emphasizing the crucial role of human body dielectric polarization effect.

## Discussion

In this study, we report a body dielectric polarization-enabled textile for ambient electromagnetic energy recovery that can be used to provide sustainable power for multiple wearable devices all around the human body. It utilizes the human body as a Gaussian surface, converting environmental EM energy into polarization energy through ion spatial polarization and dipole polarization of water molecules. Our approach addresses the energy supply problem of distributed electronic devices in body area networks. We demonstrated the use of body dielectric polarization-enabled textiles to drive a wireless heart rate monitoring belt and achieve wireless Bluetooth communication with a watch terminal. Our results highlight the potential of harnessing the design of the human body and clothing to harvest electromagnetic energy to power electronic

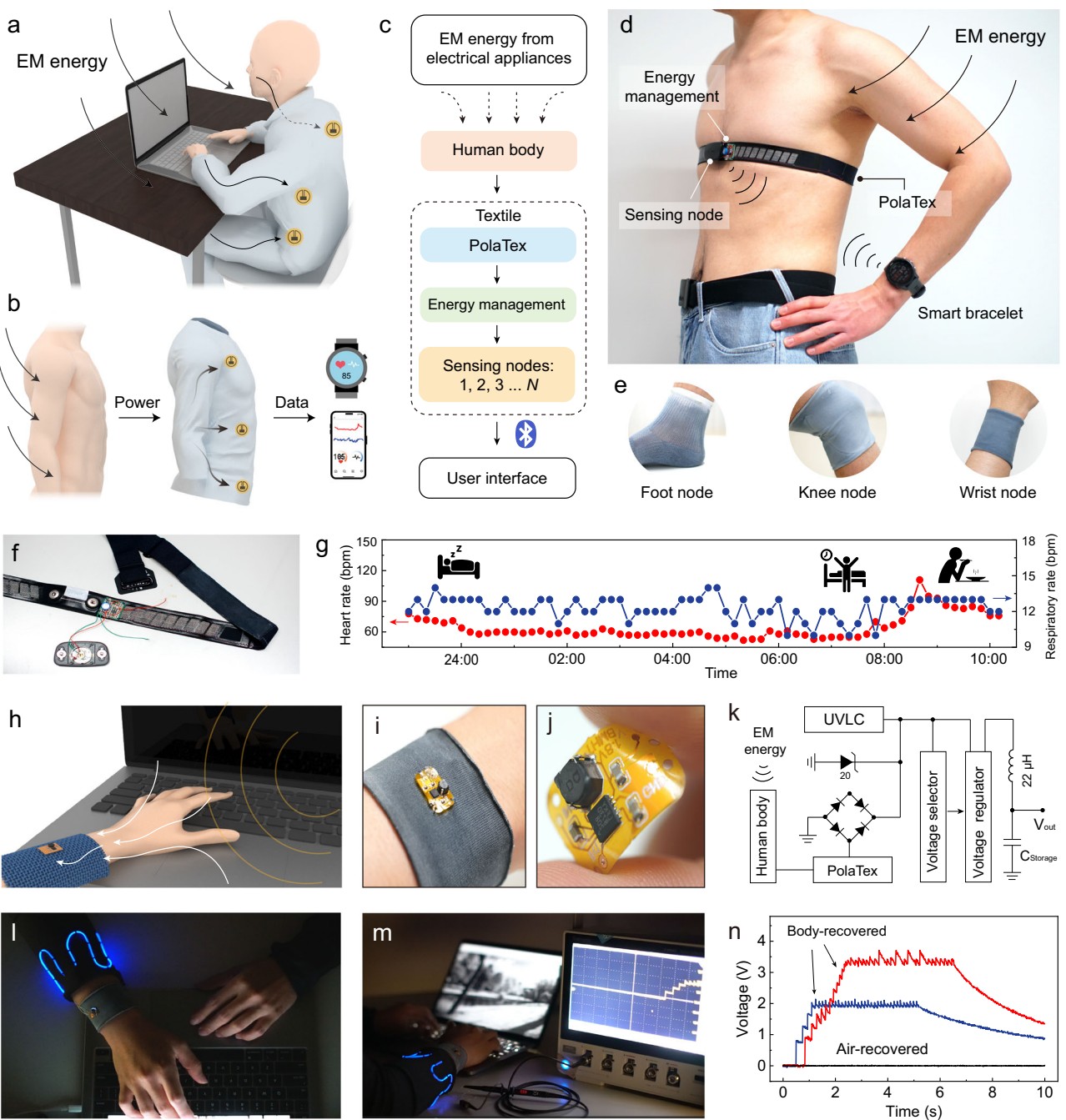

**Fig. 4 | Application of body dielectric polarization-enabled textiles in body area networks. a** Schematic diagram of the EM energy recovery application scenarios. **b** Schematic diagram of a multi-node system based on the EM energy recovery principles. **c** Energy flow pathways in the process of EM energy recovery: from the ambient energy to powering personal electronics. **d** Application of PolaTex for a wireless heart rate monitoring device. **e** PolaTex recovered EM energy at some important anatomical sites. **f** Digital photo of the heart rate band powered by the PolaTex. **g** Data of the heart rate and respiratory rate acquired from a participant during long-term monitoring using an HR belt powered entirely by the developed textile. **h** Schematic diagram of the body dielectric polarization-enabled wristband. **i** Digital photo of the body dielectric polarization-enabled wristband. **j** Photograph of the flexible energy management circuit. **k** Equivalent circuit diagram of the energy management circuit for PolaTex. UVLC: Undervoltage Lockout Circuit, making sure not to start before the voltage is sufficient. **l, m** Application of PolaTex in powering light-emitting fibers. **n** Voltage-time curves of the PolaTex when powering the electronic device, recorded while worn by different human participants, and while not worn (recovery in air).

devices, providing a starting point for a textile platform for distributed wireless body area networks.

## Methods
### Ethics statement
All tests involving human participants were considered by ETH Zurich Ethics Commission, establishing that the project is not considered human subject research and thus exempt from approval. Written and signed informed consent of all participants was obtained prior to tests and inclusion in this study.

### Materials
Poly(2,3-dihydrothieno-1,4-dioxin)-poly (styrene sulfonate) (PEDOT: PSS, 483095), Porcine Gelatin (G1890), Ammonium sulfate, and

glycerin used in this study were purchased from Sigma-Aldrich and used as received without further purification. Lycra cotton textiles were purchased from A PROPOS, Switzerland (93738, 80% cotton).

## Preparation process of the PolaTex

First, a commercial cotton textile ($10 \times 10$ cm) was soaked in 30 mL of PEDOT: PSS for 10 min to ensure full impregnation of the textile by the conductive polymer. Next, gelatin powder was dissolved in a mixture of deionized water and glycerol (mass ratio 3:2) to prepare a gelatin solution with a concentration of 15 wt%, followed by stirring at 80 °C for 30 min and ultrasonic degassing. The gelatin hydrogel solution was then used as a spray coating and uniformly deposited onto the conductive textile PEDOT: PSS surface using a standard airbrush device with compressed air as propellant. The sprayed textile samples were placed in a fridge at 4 °C for 20 min to facilitate gelation of the hydrogel. Last, the textile sample was soaked in ammonium sulfate solution (20 wt%) at room temperature for 12 h to obtain low-impedance, air-permeable PolaTex.

## Design and fabrication of energy management circuits

An ISO 9001:2015-compliant vendor (JLCPCB.com) fabricated the flexible Printed Circuit Board (fPCB) according to our design. The circuit consists of an energy management chip, LTC-3588 and other basic electronic components. The $C_1$ capacitor, as an energy storage, can be replaced according to different electrical appliances.

## Characterization and measurements

The mechanical tests were conducted using a universal tensile/compression testing machine (ZwickiLine Z1.0, Zwick Roell). The morphologies of various samples were examined using a scanning electron microscope (Phenom XL G2, Thermo Scientific). The dielectric constant of different materials was measured using an inductance-capacitance-resistance (LCR) meter via a wired connection with the materials. Contact angle studies for the developed materials were recorded using a Biolin Scientific Theta Lite optical tensiometer in sessile drop mode. Distilled water was used as a testing liquid; the initial droplet volume was 5 μL. The textile permeability to air was measured by the air permeability tester, following GB/T 24218.15-2018 standard.

## Voltage and spectrum testing of EM energy recovery

The electrical output of the PolaTex was evaluated using an oscilloscope (MDO34, Tektronix), where the voltage signal was measured by the voltage mode of the oscilloscope (10 MΩ), and the wireless signal was measured by the RF mode of the oscilloscope. As shown in Supplementary Fig. 22, we provide detailed methodology for voltage and spectrum measurements of EM energy recovering using PolaTex. A 1 MΩ internal resistance was set on the oscilloscope, with the positive terminal of the probe connected to PolaTex and the ground terminal left floating. PolaTex was fixed onto the wrist during measurement. Both the voltage and spectrum modes of the oscilloscope were employed to capture the recovered electrical signals. To minimize EM interference from surrounding electrical devices, all appliances within a 2-meter radius were removed. Subsequently, the distance between the human body and the emission source (a laptop) was gradually adjusted to collect recovered electrical signals under varying radiation intensities.

## Skin-electrode interface impedance measurement

Impedance characteristics were measured using a potentiostat (μStat-i400, MetrOhm DropSens) in a three-electrode configuration (Supplementary Fig. 12). A three-electrode configuration was employed using an electrochemical workstation to measure the interface impedance over a frequency range of 1 Hz to 1 MHz. The total measured impedance comprises the electrode impedance ($Z_{WE}$), skin impedance ($Z_{skin}$), and the interface impedance ($Z_{intWE}$). PolaTex was used as the working electrode, while commercial hydrogel electrodes (3 M) served as both reference and counter electrodes. The distance between the working and counter electrodes was set to 10 cm, and the distance between the working and reference electrodes was maintained at 1 cm.

## Data availability

Relevant data supporting this study are available within the article and the Supplementary Information file. All the numerical data generated in this study are provided in the Supplementary Information/Source Data file in the form of an Excel file with pages corresponding to each presented graph. All data, including images, are available from the corresponding author upon request. Source data are provided with this paper.

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

## Acknowledgments

This work was supported by ETH Zurich. We acknowledge the discussion with the BHMT group members. We sincerely appreciate Shuaixin Qi for his participation and assistance in this project. We appreciate Tharme Lewis for his assistance with this project.

## Author contributions

W.Y., Y.L., and C.M. conceived the project and designed the experiments. W.Y. and Y.L. fabricated the samples and devices, ran the experiments, analyzed the data, and wrote the initial draft of the manuscript. A.S. carried out some experiments and interpretation of the results, W.Y., A.S, and C.M. reviewed the manuscript. W.Y. and C.M. supervised the project.

## Funding

## Competing interests

The authors declare no competing interests.
