## [Peer Review file · Nature Communications]

Leveraging body dielectric polarization for ambient electromagnetic energy recovery via e-textile

Corresponding Author: Dr Weifeng Yang

Version 0:

Reviewer comments:

Reviewer #1

(Remarks to the Author)

The submitted manuscript titled "PolaTex: leveraging body dielectric polarization for ambient electromagnetic energy recovery via e-textile" presents a novel body dielectric polarization-enabled textile for electromagnetic energy recovery. While the topic is of significant interest, some issues need to be resolved to clarify the claims that are made by the authors:

1. Line 83, the authors mentioned that the vision of smart clothing includes the washable property. Have the authors tested the washable performance of the proposed PolaTex to conform to this vision?
2. The key step of EM energy recovery depends on charge accumulation in human body based on capacitive effect between the air and the human body. Since the physical condition is different in different bodies, the authors need to evaluate the performance of PolaTex on different bodies to explore the universality.
3. According to the authors' description, Leica cotton textile is the flexible substrate, PEDOT:PSS provides the conductivity, and the gelatin provides a low impedance. Can the proposed PolaTex be considered as a flexible electrode on the skin? If so, will typical textile-based flexible electrodes (e.g. textiles made with silver fabrics) exhibit better performance?
4. What is the role of glycerol in the hydrogel?
5. Generally, hydrogels are at risk of drying out after being placed in the environment for a period of time. Given this, how durable is the proposed PolaTex?
6. Ammonium sulfate has an irritating effect on the skin. Is there any potential health hazard of the PolaTex treated with the ammonium sulfate?

Reviewer #2

(Remarks to the Author)

The authors propose a very interesting approach for harvesting electromagnetic energy in environments. By leveraging space charge and dipole polarization effects within the human body, the functional textile developed in this work is capable of directly converting electromagnetic energy from the skin surface without the need for additional power-generating components. The reported energy output, up to 10.2 mW/person from a common laptop during routine typing, is indeed impressive and demonstrates potential for real-world application. However, the present study primarily introduces a concept, without sufficient discussion or validation of its practical feasibility. As a wearable textile, the proposed PolaTex system must also demonstrate essential characteristics such as washability, comfort, long-term stability, and user safety, which are either insufficiently addressed or entirely omitted in the current version. Therefore, I do not recommend this manuscript for publication at this stage. However, I hope the following comments will be helpful for future revision.

1. The authors mention that the vision of smart clothing includes being "comfortable and washable." Accordingly, the mechanical and chemical stability of the PolaTex system should be thoroughly evaluated. This includes assessing the electrical performance of the PEDOT:PSS and hydrogel components under repeated mechanical deformation (e.g., bending and stretching) as well as after multiple washing cycles. The impact of these factors on the overall power conversion efficiency should also be quantified.
2. Considering that the PolaTex system needs to operate in direct contact with human skin, biocompatibility and skin irritation assessments are essential to ensure long-term user safety. The absence of such tests raises concerns regarding the viability of the system for practical wearable applications.
3. The manuscript contains an excessive number of schematic illustrations (e.g., Figures 1a,b; 2c,d,f; 3a,j,i; 4a,b). While

these visuals enhance the aesthetic appeal, they do not contribute substantively to its scientific importance. The authors are encouraged to include more data that highlight material innovations and provide in-depth device and system characterization.

4. COMSOL simulation, experimental methods, and circuit design should be presented in greater detail to ensure reproducibility.

5. In Line 21, the authors state that the textile can recover "up to 10.2 mW/person (10.01 dBm)" from a common laptop during daily typing." However, in Line 214, they report a value of "up to 6.05 mW/person." The discrepancy between these two values must be clarified. Additionally, the calculation methodology for these figures should be explicitly described. For instance, does "10.2 mW/person" assume that the entire body surface is covered by the functional textile?

6. The manuscript text and figures should be carefully reviewed for clarity and consistency. For instance: 1) Line 58 contains a typo error: "thus can only be sued for..."; 2) Line 214 refers to "as shown in xx," which appears to be an incomplete placeholder; 3) In Figure 1g, the PolaTex system should be clearly displayed rather than focusing on the laptop. Moreover, Figure 1g seems to convey the same information as Figures 4h, i, j, and m, resulting in unnecessary redundancy; to name a few.

Reviewer #3

(Remarks to the Author)

This study reported a body dielectric polarization-enabled e-textile for ambient EM energy recovery that can provide sustainable power for multiple wearable devices all around the human body. By leveraging the space charge/dipole polarization within the body, the e-textile can directly convert EM energy from the skin's surface, eliminating the need for additional power-generation components. This novel approach effectively tackles the energy supply challenge for distributed electronic devices in body area networks. It offers a fresh perspective on powering body area networks and is anticipated to drive the advancement of intelligent wearable devices and body area networks. Overall, the research is both innovative and practical. However, there are still aspects that need to be refined. The following are some specific details.

1. Considering the e-textile's contact with the skin, what is its biocompatibility?
2. When e-textiles soaked in PEDOT: PSS solutions of different concentrations recover EM energy, is there a correlation between the power density and the concentration of the soaking solution?
3. In S9, the pictures are mislabeled.
4. In S14, why is there a minimum value under different concentration loads?
5. What is the long-term working stability of the e-textile? Can it operate properly under conditions of human sweating?
6. How stable is the e-textile under different environmental conditions, such as high temperature and high humidity?
7. Does the long-term utilization of human body dielectric polarization for harvesting ambient electromagnetic energy have potential impacts on human health?

Version 1:

Reviewer comments:

Reviewer #1

(Remarks to the Author)

Comments have been well addressed and explained in revised manuscript. I therefore think the manuscript has met the requirements for publication now.

Reviewer #2

(Remarks to the Author)

The authors have thoroughly addressed all of my previous comments. I am satisfied with the revisions and have no further concerns.

Reviewer #3

(Remarks to the Author)

The authors have thoroughly and meticulously addressed all the comments I raised during the review process. They have conducted extensive additional experiments and made in-depth and comprehensive revisions to the manuscript from multiple perspectives, including user comfort, safety, and environmental stability. These targeted modifications have significantly improved the overall quality and academic rigor of the paper. The revisions reflect the authors' rigorous scientific attitude and their high level of respect for the reviewers' feedback. Given the substantial improvements in the manuscript's content, structure, and clarity, as well as the authors' constructive and comprehensive responses to all reviewer comments, I believe the current version meets the publication standards of Nature Communications and recommend acceptance.

Response Letter to Reviewers

We would like to show our sincere appreciation to the reviewers for their time and valuable comments on our manuscript "*PolaTex: leveraging body dielectric polarization for ambient electromagnetic energy recovery via e-textile*" (NCOMMS-25-21928-T). The manuscript has been carefully revised in light of the reviewers' comments and suggestions. All the changes in the manuscript and supporting information made in this revision are highlighted in yellow in the "Revised Manuscript" and "Revised Supporting Information", respectively. A detailed point by point response to the comments and suggestions is provided below.

Reviewer #1:

This submitted manuscript titled "PolaTex: leveraging body dielectric polarization for ambient electromagnetic energy recovery via e-textile" presents a novel body dielectric polarization-enabled textile for electromagnetic energy recovery. While the topic is of significant interest, some issues need to be resolved to clarify the claims that are made by the authors:

Response: We are grateful to the reviewer for all the valuable comments and suggestions, which have helped to improve the quality of this manuscript significantly. You will find our replies to the questions point by point below:

1) Line 83, the authors mentioned that the vision of smart clothing includes washable property. Have the authors tested the washable performance of the proposed PolaTex to conform to this vision?

Response: Thanks for your suggestion, washability is indeed extremely important for the e-textiles. We have further evaluated the washability of PolaTex through 10 washing cycles. As illustrated in Fig. R1a, 3 mL of laundry detergent and 1 liter of deionized water were added to a glass container along with the PolaTex sample. Magnetic stirring at 600 rpm was applied at room temperature to simulate a typical washing environment. Then PolaTex was immersed directly into the solution without any protective packaging. Each wash cycle lasted 30 minutes. After washing, the sample was air-dried naturally before conducting electrical output measurements. We assessed the electromagnetic energy recovery performance of PolaTex before washing and after the 1st, 2nd, 4th, and 10th wash cycles. The results showed no significant degradation in the electrical output signals. Additionally, microscopic imaging was used to observe the surface morphology of PolaTex after the wash cycles. The hydrogel layer remained well-adhered to the electronic textile, indicating good structural integrity. These results collectively demonstrate that PolaTex possesses notable resistance to washing.

● **Our revision to the manuscript:**

We added Fig. R1 as Fig. S14. We supplemented the description of washing performance and washing process in the manuscript Methods part. Relevant modifications have been highlighted in red within the revised manuscript and supplementary materials.

Fig. R1 Washability of PolaTex. (a) Digital photo of the Pola-Tex washing process. (b) EM energy recovery performance of Pola-Tex before and after washing. (c) Microscopic morphology of the hydrogel layer on the surface of Pola-Tex before and after washing.

2) The key step of EM energy recovery depends on charge accumulation in human body based on capacitive effect between the air and the human body. Since the physical condition is different in different bodies, the authors need to evaluate the performance of PolaTex on different bodies to explore the universality.

Response: Thanks for your advice. In fact, in our initial manuscript, we have already measured the output electrical signals from various body parts and different number of collection channels in multiple individuals, as shown in the manuscript of Figs. 3j-m. In order to make this aspect of the

study clearer, we now supplement digital photographs showing the EM energy signals recovered by PolaTex from different Participants (Fig. R2).

- **Our revision to the manuscript:**

We added Fig. R2 as Fig. S24. Relevant modifications have been highlighted in red within the revised supplementary materials.

Fig. R2 Digital photos of different participants recovering EM energy through PolaTex.

3) According to the authors' description, Leica cotton textile is the flexible substrate, PEDOT:PSS provides the conductivity, and the gelatin provides a low impedance. Can the proposed PolaTex be considered as a flexible electrode on the skin? If so, will typical textile-based flexible electrodes (e.g. textiles made with silver fabrics) exhibit better performance?

Response: Thank you for your question. As you have noted, in our PolaTex system, a hydrogel layer is incorporated to reduce the skin–electrode interface impedance, thereby enhancing the efficiency of EM energy recovering. While the intrinsic resistance of the conductive fabric ($\sim 10^2 \Omega$) and the hydrogel ($\sim 10^3 \Omega$) contribute marginally to energy loss, the skin–electrode interface impedance (on the order of $\sim 10^5 \Omega$) is a far more critical factor affecting overall performance.

In order to assess how PolaTex performs as a flexible electrode as compared to commercial ones, we compared the EM energy recovering capabilities of PolaTex with those of commercial silver-plated fabric electrodes under identical test conditions, as shown in Fig. R3a. PolaTex achieved a recovered voltage of 7.5 V, whereas the silver-plated fabric electrodes reached only around 4 V. This disparity arises because commercial silver-plated electrodes, as dry electrodes, often exhibit high skin–electrode interface impedance (typically $\sim 10^6 \Omega$ at 50 Hz) [*J. Micromech. Microeng.* 2011, 21(8): 085014], due to poor conformal contact with the skin despite their low intrinsic resistance. By contrast, PolaTex achieves a much lower skin–electrode interface impedance of approximately $5 \times 10^4 \Omega$ at 50 Hz (Fig. S12), enabling significantly improved voltage recovery and overall efficiency.

Fig. R3 (a) Digital photos and (b) voltage data comparing EM energy recovery from commercial textile electrodes and PolaTex.

4) What is the role of glycerol in the hydrogel?

Response: Thank you for your professional suggestion. The incorporation of glycerol into the hydrogel layer serves to enhance the water retention capability of gelatin-based hydrogels. In our initial manuscript, we systematically investigated the water retention performance of hydrogels with varying water-to-glycerol mass ratios. We have now performed additional measurements in regards to this effect. As shown in Fig. R4, the gelatin hydrogel without glycerol loses nearly all of its water content within 20 hours. In contrast, water retention significantly improves as the glycerol content increases. Notably, when the water-to-glycerol mass ratio reaches 10:6, the hydrogel retains more than 80 wt% of its original mass even after 80 hours at ambient conditions in air.

Fig. R4. Water retention test of PolaTex with different water/glycerol ratios.

5) Generally, hydrogels are at risk of drying out after being placed in the environment for a period of time. Given this, how durable is the proposed PolaTex?

Response: Indeed, water loss in hydrogels remains a well-known challenge in this field. To mitigate this issue, we have implemented the following strategies:

Fig. R5 (a) Optical microscopy and (b) digital photographs of hydrogel dehydration and its water recovery properties.

(1) Incorporation of glycerol to improve its water retention: Glycerol was introduced into the hydrogel system to improve its water retention capability. As a result, the hydrogel was able to maintain a stable hydrated state for 2 to 3 days (Fig. R4), as we discussed in the last question.

(2) Rehydration via immersion in a water environment: We further conducted rehydration tests on dehydrated hydrogels. Due to the physically cross-linked nature of our gelatin-based hydrogel, water can readily re-enter the dehydrated network (Fig. R5a-2, R5b-2). As shown in Fig. R5,

after immersing the dehydrated PolaTex in deionized water at 30 °C for 30 minutes, the hydrogel was able to reabsorb water and restore its original structure (Fig. R5a-3, R5b-3). Furthermore, we carried out a second cycle of dehydration and rehydration (Fig. R5a-4,5 and R5b-4,5) and found that the hydrogel layer could still effectively recover its water retention properties after repeated cycling.

- **Our revision to the manuscript:**

We added Fig. R5 as Fig. S15. Relevant modifications have been highlighted in red within the revised supplementary materials.

REDACTED

Thank you again for your valuable comments and suggestions!

Reviewer #2:

The authors propose a very interesting approach for harvesting electromagnetic energy in environments. By leveraging space charge and dipole polarization effects within the human body, the functional textile developed in this work is capable of directly converting electromagnetic energy from the skin surface without the need for additional power-generating components. The reported energy output, up to 10.2 mW/person from a common laptop during routine typing, is indeed impressive and demonstrates potential for real-world applications. However, the present study primarily introduces a concept, without sufficient discussion or validation of its practical feasibility. As a wearable textile, the proposed PolaTex system must also demonstrate essential characteristics such as washability, comfort, long-term stability, and user safety, which are either insufficiently addressed or entirely omitted in the current version. Therefore, I do not recommend this manuscript for publication at this stage. However, I hope the following comments will be helpful for future revision.

Response: We are thankful for the reviewer's discussion and proposals. They were very helpful in improving the manuscript in many aspects. We provide detailed answers to every question below:

1) The authors mention that the vision of smart clothing includes being "comfortable and washable." Accordingly, the mechanical and chemical stability of the PolaTex system should be thoroughly evaluated. This includes assessing the electrical performance of the PEDOT:PSS and hydrogel components under repeated mechanical deformation (e.g., bending and stretching) as well as after multiple washing cycles. The impact of these factors on the overall power conversion efficiency should also be quantified.

Response: Thank you for your valuable suggestions, indeed washability is a paramount property for any e-textiles. Regarding the requirements, we have quantitatively characterized the washability and mechanical cyclic stability of PolaTex in the following tests.

(1) Washability of the PolaTex after 10 washing cycles : As illustrated in Fig. R8a, 3 mL of laundry detergent and 1 liter of deionized water were added to a glass container along with the PolaTex sample. Magnetic stirring at 600 rpm at room temperature was applied to simulate a typical washing environment. Then PolaTex was immersed directly into the solution without any protective packaging. Each wash cycle lasted 30 minutes. After washing, the sample was air-dried naturally before conducting electrical output measurements. We assessed the electromagnetic energy recovering performance of PolaTex before washing and after the 1st, 2nd, 4th, and 10th wash cycles. The results showed no significant degradation in the electrical output signals. Additionally, microscopic imaging was used to observe the surface hydrogel morphology of PolaTex after various wash cycles. The hydrogel layer remained well-adhered to the electronic

textile, indicating good structural integrity. These results collectively demonstrate that PolaTex possesses notable resistance to washing.

Fig. R8 Washability of the PolaTex. (a) Digital photo of the Pola-Tex washing process. (b) EM energy recovery performance of Pola-Tex before and after washing. (c) Microscopic morphology of the hydrogel layer on the surface of PolaTex before and after washing.

(2) **Mechanical cyclic stability of the PolaTex:** We performed 1000 cycles of bending and compression tests (Fig. R9a). As shown in Fig. R8b, the mechanical performance of Pola-Tex exhibited no significant degradation throughout the 1000 bending cycles, which indirectly confirms the strong interfacial adhesion between the hydrogel layer and the cotton fabric substrate. Furthermore, the ability of Pola-Tex to recover environmental electromagnetic energy remained stable, with no notable decline in performance before and after the mechanical cycling (Figs. R9c and d).

Fig. R9 Mechanical cyclic stability of the PolaTex. (a) Digital photo of the PolaTex bending cycle test process. **(b)** Mechanical data of PolaTex bending cycle under 1000 cycles. **(c)** Electrical output performance of PolaTex after 1000 bending cycles. **(d)** Electrical output performance of PolaTex after 1000 compression cycles.

- **Our revision to the manuscript:**

We added Fig. R8, R9 as Fig. S14 and S18. Relevant modifications have been highlighted in red within the revised supplementary materials.

REDACTED

3) The manuscript contains an excessive number of schematic illustrations (e.g., Figures 1a,b; 2c,d,f; 3a,j,i; 4a,b). While these visuals enhance the aesthetic appeal, they do not contribute substantively to its scientific importance. The authors are encouraged to include more data that highlight material innovations and provide in-depth device and system characterization.

Response: Thanks for your professional suggestions. We have made appropriate revisions to the figures in the manuscript by including more data and characterization results in the figures of the

main text. Specifically, we have incorporated the **impedance data of different concentrations of salt solutions** at various frequencies from the Supplementary Information into Fig. 2f of the main manuscript. Additionally, we have supplemented the study on the **effect of PolaTex area on environmental EM energy recovery** (Figs. 3 h and i). Furthermore, we have added the **quantitative cytotoxicity data of PolaTex** (Fig. S27). Relevant modifications have been highlighted in red within the revised manuscript and supplementary materials.

4) COMSOL simulation, experimental methods, and circuit design should be presented in greater detail to ensure reproducibility.

Response: Thank you for your suggestion. We have significantly extended the parts of the manuscript in question. In the revised manuscript, we provide the following detailed explanations of COMSOL simulation, experimental methods, and circuit design.

Fig. R12 COMSOL simulation parameter settings

(1) COMSOL Simulation Setup

To investigate the interaction between low-frequency electromagnetic fields and the human body, we performed simulations using *COMSOL Multiphysics* with the “**Electric Currents**” physics interface. This module allows us to account for both ionic space charge polarization and molecular dipole polarization, which are crucial for accurately modeling the combined effects of electrical conductivity and permittivity in biological tissues and surrounding media. Both cross-sectional and 3D field plots were generated to visualize the spatial distribution of the field around and inside the body (see Fig. R12).

1. Model Selection: Electric Currents (EC module); Frequency Domain (50 Hz)

2. Geometry: A 3D human body model was imported into COMSOL to represent a standing individual. A point source transmitter was placed at a fixed distance in front of the body to simulate a low-frequency electric field emission source (100 V, 50 Hz).

3. Material properties: **Air:** $\epsilon = 1$ at 50 Hz; $\sigma = 1 \times 10^{-14}$ S/m; **Human body:** $\epsilon = 1 \times 10^6$ at 50 Hz ;
 $\sigma = 0.1$ S/m

4. Boundary and source settings: The transmitter was set as an electric potential boundary with 100 V amplitude and 50 Hz sinusoidal excitation. All external boundaries were set to electrical insulation (zero normal current flow).

5. Mesh settings: A **fine tetrahedral mesh** was applied, particularly around the body–air interface and the source region, to ensure spatial resolution of electric field gradients.

6. Postprocessing: Electric potential distribution (V) and Electric field intensity (E-field, V/m) throughout the human body and surrounding air.

(2) Experimental methods:

Skin-electrode interface impedance measurement: As shown in Fig. S12, a three-electrode configuration was employed using an electrochemical workstation to measure the interface impedance over a frequency range of 1 Hz to 1 MHz. The total measured impedance comprises the electrode impedance (Z_{WE}), skin impedance (Z_{skin}), and the interface impedance (Z_{intWE}). PolaTex was used as the working electrode, while commercial hydrogel electrodes (3M) served as both reference and counter electrodes. The distance between the working and counter electrodes was set to 10 cm, and the distance between the working and reference electrodes was maintained at 1 cm.

Voltage and spectrum testing of EM energy recovery based on PolaTex: As shown in Fig. R13, we provide the detailed methodology for voltage and spectrum measurements of EM energy recovering using PolaTex. A 1 M Ω internal resistance was set on the oscilloscope, with the positive terminal of the probe connected to PolaTex and the ground terminal left floating. PolaTex was fixed onto the wrist during measurement. Both the voltage and spectrum modes of the oscilloscope were employed to capture the recovered electrical signals. To minimize EM interference from surrounding electrical devices, all appliances within a 2-meter radius were removed. Subsequently, the distance between the human body and the emission source (a laptop) was gradually adjusted to collect recovered electrical signals under varying radiation intensities.

(3) Design and fabrication of energy harvesting circuit

An ISO 9001:2015-compliant vendor (JLCPCB.com) fabricated the flexible Printed Circuit Board (fPCB) according to our design. The circuit consists of an energy management chip LTC-3588 and other basic electronic components. The C_1 capacitor, as an energy storage, can be replaced according to different electrical appliances.

Fig. R13 Voltage and spectrum testing of EM energy recovery based on PolaTex. (a-d)
Different distances between the human body and the transmitter.

● **Our revision to the manuscript:**

We added Fig. R13 as Fig S22. We have significantly expanded the methods section with detailed experimental methods in the revised manuscript. Relevant modifications have been highlighted in red within the revised supplementary materials.

5) In Line 21, the authors state that the textile can recover "up to 10.2 mW/person (10.01 dBm)" from a common laptop during daily typing." However, in Line 214, they report a value of "up to 6.05 mW/person." The discrepancy between these two values must be clarified. Additionally, the calculation methodology for these figures should be explicitly described. For instance, does "10.2 mW/person" assume that the entire body surface is covered by the functional textile?

Response: Thank you for your observation. Indeed, it is hard to compare these values, and it was not very clear why in the previous version of the manuscript. In the revised text, we explain the differences in power density and the recovery of EM energy with different PolaTex areas as follows.

(1) **The discrepancy between two power density values:** Since there is no unified power calculation standard for this new technology based on human body dielectric polarization to recover electromagnetic energy, different calculation methods will lead to different results of the final power density. In addition, as shown in Fig. 3h, differences in skin surface conditions between

different individuals can also lead to differences in the voltage results of the test recovery, which in turn leads to different power densities. To avoid ambiguity, we have additionally annotated the calculation method for the output power density in the text.

The power density converted in RF mode: 10.2 mW/person is obtained by converting the peak power of 10.01 dBm at 50 Hz in Fig. 1f through the formula (1):

$$P_w = 10^{\frac{dBm-30}{10}} \quad (1)$$

Where P_w is the power density recovered by PolaTex, and dBm is the decibel relative to one milliwatt which was obtained using the RF mode of an oscilloscope.

Power density is calculated from voltage: 6.05 mW/person was calculated by the formula (2):

$$P_w = \frac{U^2}{R} \quad (2)$$

Where P_w is the power density recovered by PolaTex; U is the recovered voltage by PolaTex; R is the external load resistor.

Thus, while these two calculations reflect the same parameter (power), they are different because they use different inputs due to different experimental conditions (and what is being measured).

(2) Effect of different individual sizes and PolaTex areas on the output signal: As shown in Fig. R14, in this dielectric polarization-based EM energy recover technology, the individual size of the dielectric will significantly affect the recovery power. This phenomenon was already investigated in our initial manuscript. Ultimately, the exposed Gaussian surface area in the EM environment directly affects the polarization area, thus leading to different output signals. In this revision, we further examined the effect of Pola-Tex's surface area on the EM energy recovery (Fig. R14b). The results show that increasing the electrode area from **0.2 cm² to 100 cm²** does not lead to a significant increase in the recovered electromagnetic voltage (Fig. R14c). We speculate that this is due to the fact that the skin–electrode interface impedance also increases with larger electrode areas, thereby counteracting the benefits of the expanded polarization area and resulting in no significant change in the final output electrical signal.

Fig. R14 Effect of different (a) individual Gaussian body sizes and (b, c) PolaTex areas on the output signal.

● **Our revision to the manuscript:**

We added Fig. R14 b and c as Fig 3h and i. Relevant modifications have been highlighted in red within the revised manuscript and supplementary materials.

6) The manuscript text and figures should be carefully reviewed for clarity and consistency. For instance: 1) Line 58 contains a typo error: "thus can only be sued for..."; 2) Line 214 refers to "as shown in xx," which appears to be an incomplete placeholder; 3) In Figure 1g, the PolaTex system should be clearly displayed rather than focusing on the laptop. Moreover, Figure 1g seems to convey the same information as Figures 4h, i, j, and m, resulting in unnecessary redundancy; to name a few.

Response: Thank you very much for your suggestions and edits. We corrected the spelling errors in Line58 and Line214 in the initial manuscript. We replaced Fig. 1g with a more detailed photo of the PolaTex system. We deleted Fig. 4k in the initial manuscript to reduce redundancy, as you suggested, and added a circuit diagram of the energy management fPCB. Relevant modifications have been highlighted in red within the revised manuscript and supplementary materials.

Thank you again for your valuable comments and suggestions!

Reviewer #3:

This study reported a body dielectric polarization-enabled e-textile for ambient EM energy recovery that can provide sustainable power for multiple wearable devices all around the human body. By leveraging the space charge/dipole polarization within the body, the e-textile can directly convert EM energy from the skin's surface, eliminating the need for additional power-generation components. This novel approach effectively tackles the energy supply challenge for distributed electronic devices in body area networks. It offers a fresh perspective on powering body area networks and is anticipated to drive the advancement of intelligent wearable devices and body area networks. Overall, the research is both innovative and practical. However, there are still aspects that need to be refined. The following are some specific details.

Response: We thank the reviewer for their insightful comments and constructive suggestions, which have greatly contributed to enhancing this manuscript. Our detailed responses to each point are provided below:

REDACTED

2) When e-textiles soaked in PEDOT: PSS solutions of different concentrations recover EM energy, is there a correlation between the power density and the concentration of the soaking solution?

Response: Indeed, this is an interesting question. As per your suggestion, we further studied the effect of different concentrations of PEDOT:PSS solution (Fig. R17a) on PolaTex electromagnetic energy recovery. As the concentration of PEDOT:PSS decreases, its resistance increases significantly from ~20 K to ~5 MΩ (Fig. R17b), but its voltage attenuation for the final recovered electromagnetic

energy is not significantly affected(Fig. R17c). This is mostly likely due to the fact that as long as electrical contact with skin is maintained by the flexible electrode, energy recovery remains possible.

Fig. R17 Relationship between PEDOT: PSS immersion concentration and output performance in PolaTex. (a) digital photo of PolaTex with different concentration of PEDOT:PSS on human arm; **(b)** resistance of the PolaTex with different weight% of PEDOT: PSS; **(c)** EM energy recovery by the PolaTex with different PEDOT: PSS loads.

3) In S9, the pictures are mislabeled.

Response: Thank you for your find. We have added fixed the label of the scale bar at Fig. S9.

4) In S14, why is there a minimum value under different concentration loads?

Response: It seems that the difference in values was in error. We apologize for the earlier test results, where a 5% reduction in stress was observed, likely due to experimental errors and testing variability. We have now repeated stress tests, and the updated results are presented in Fig. R18.

Fig. R18 The stress-time curves of hydrogels with different PEDOT: PSS doping concentrations.

- **Our revision to the manuscript:**

We revised Fig. R14 b as Fig. S18a. Relevant modifications have been highlighted in red within the revised supplementary materials.

5) What is the long-term working stability of the e-textile? Can it operate properly under conditions of human sweating?

Response: Thank you for your comment. Those are indeed important questions for any e-textile.

- (1) Long-term working stability:** Most parts of the PolaTex are quite robust and stable, with the hydrogel being the weakest point stability-wise. Indeed, water loss in hydrogels remains a well-known challenge in this field. To mitigate this issue, we have implemented the following strategies.
Rehydration via immersion in a water environment: We further conducted rehydration tests on dehydrated hydrogels. Due to the physically cross-linked nature of our gelatin-based hydrogel, water can readily re-enter the dehydrated network (Fig. R19a-2, R19b-2). As shown in Fig. R19, after immersing the dehydrated PolaTex in deionized water at 30 °C for 30 minutes, the hydrogel was able to reabsorb water and restore its original structure (Fig. R19a-3, R19b-3). Furthermore, we carried out a second cycle of dehydration and rehydration (Fig. R19a-4, 5 and R19b-4, 5) and found that the hydrogel layer could still effectively recover its water retention properties after repeated cycling.
Incorporation of glycerol to improve its water retention: Glycerol was introduced into the hydrogel system to improve its water retention capability. As a result, the hydrogel was able to maintain a stable hydrated state for 2 to 3 days (Fig. R21a).
- (2) Mechanical cyclic stability of the PolaTex.** Apart from hydrogel stability, we have also performed long-term cyclical bending and wear tests on PolaTex, which show that the performance of the textile is not affected by neither of these factors (Fig. R20).
- (3) Operating under sweating state:** We used a phosphate-buffered saline (PBS) solution with pH 7 to simulate sweat and applied it to the skin surface (Fig. R21b). The electrical output performance of PolaTex was compared before and after exposure to PBS (Fig. R21c). The results demonstrated that PolaTex remained stable under simulated sweating conditions and even exhibited a slight increase in harvested environmental electromagnetic (EM) energy, with the peak-to-peak voltage (V_{pp}) increasing from 7.3 V to 7.8 V. This enhancement is likely attributed to the presence of ions in the sweat, which may reduce the skin–electrode interface impedance and facilitate improved energy transfer. This indicates that sweating would be beneficial to the performance of the system.

Fig. R19 (a) Optical microscopy and (b) digital photographs of hydrogel dehydration and its water recovery properties.

Fig. R20 Mechanical cyclic stability of the PolaTex. (a) Digital photo of the PolaTex bending cycle

test process. (b) Mechanical data of PolaTex bending cycle under 1000 cycles. (c) Electrical output performance of PolaTex after 1000 bending cycles. (d) Electrical output performance of PolaTex after 1000 compression cycles.

Fig. R21 (a) Water retention test of PolaTex with different water/glycerol ratios. (b) Digital photo of PBS simulating sweat dripping on the skin surface. (c) Comparison of EM energy recovery by PolaTex before and after adding PBS simulated sweat.

- **Our revision to the manuscript:**

We added Figs. R19, R20 and Figs. 21c, and c as Figs. S15, S18 and Fig. 19. Relevant modifications have been highlighted in red within the revised supplementary materials.

6) How stable is the e-textile under different environmental conditions, such as high temperature and high humidity?

Response: Those are important factors for wearable devices as well. We tested the changes in EM energy recovery by PolaTex in the conditions with increased temperature and increased humidity.

(1) Effect of high temperature: We used a hot air gun to simulate localized high-temperature effects in the skin microenvironment (Figs. R22a, b) and monitored the real-time changes in the EM energy recover performance of PolaTex during the heating process. As the surface temperature of PolaTex increased from 28.7 °C (Fig. R22a-1) to 34.1 °C, with local hotspots reaching up to 60 °C (Fig. R22a-4), the voltage output for EM energy harvesting remained stable at approximately 7.6 V.

(2) Effect of high humidity: We used a humidifier to simulate localized high-humidity conditions in the skin microenvironment (Fig. R22d). The results showed that as the humidity gradually increased, the peak-to-peak voltage (V_{pp}) of the EM energy recovering by PolaTex slightly increased, from 7.3 V to approximately 7.8 V (Fig. R22e). We speculate that this enhancement may be due to a reduction in the skin–electrode interface impedance or an increase in the dielectric

constant of the surrounding air under high-humidity conditions.

Fig. R22 Effect of (a-c) High temperature and (d, e) high humidity on electrical performance of PolaTex.

- **Our revision to the manuscript:**

We added Fig. R22 as Fig. S20. Relevant modifications have been highlighted in red within the revised supplementary materials.

7) Does the long-term utilization of human body dielectric polarization for harvesting ambient electromagnetic energy have potential impacts on human health?

Response: Thank you for this question. We believe that the impact of EM energy in our daily life environment on the human body can be mainly divided into (1) Joule heating effect and (2) electromagnetic radiation effect. The main analysis is as follows:

(1) **Impact of Joule heating effect:** The EM field frequency used in our study was in the low-frequency range (50 Hz), with a maximum power output of approximately 10 dBm (~10.05 mW).

At this power level, significant Joule heating effects on the human body are unlikely to occur.

(2) **Impact of EM radiation effect:** In our research, PolaTex was used to recover extremely low-

frequency EM fields at 50 Hz. Throughout the experiments, no additional signal transmitter was introduced to enhance the EM signals; only the naturally occurring EM fields emitted by daily electrical appliances were collected. Typically, the operating voltage of common household devices is around 200 V, such as 230 V/50 Hz in Switzerland. According to the standards set by the International Commission on Non-Ionizing Radiation Protection (ICNIRP), the permissible occupational exposure limit for a 50 Hz EM field is 5 kV/m [*Health Phys.*, 2010, 99: 818–836]. In other words, the EM environment surrounding the human body in our experiments was far below the 5 kV/m exposure threshold. On the other hand, several studies have specifically investigated the potential biological effects of low-frequency electromagnetic fields on human health ([1] *Front. Neurosci.*, 2023, 17: 1247021; [2] *Plos One*, 2013, 8(9): e72944; [3] *Mutat. Res. Genet. Toxicol. Environ. Mutagen.*, 2005, 583(2): 184–188). A summary of representative research findings is provided in the table below.

Table R1 Summary of research on the potential impact of low-frequency electromagnetic fields on human health

Ref	Aspects of impact	Experimental conditions	Animal species	Main effect
[1]	Immune	50 Hz 100 μT for 20 h/day for 12 weeks	Mice	No significant effects on peripheral hematopoietic system
[2]	Immune	50 Hz 7 mT 24 h	Male mice	Significant increase of blood parameters, such as white blood cells, lymphocytes, hemoglobin, and hematocrit levels.
[3]	Cell reproduction	50 Hz 25 μT for 18 weeks	Male mice	Increase in serum levels of male luteinizing hormone after 18 weeks; Decreased in testosterone levels after 6 and 12 weeks
[4]	Cell reproduction	60 Hz 8.8 μT for 72 h	Mice	Low sperm counts for 72 h exposed animals, without altering male germ cell morphological characteristics.
[5]	Cancer	50 Hz 2mT	Mice	No obviously association
[6]	Cancer	60 Hz 2mT for 29 weeks	Mice	No obviously association
[7]	Cardiovascular	50 Hz 1μT	Mice	No obviously association
[8]	Cardiovascular	60 Hz 2.4mT for 2h	Mice	The reduction of glutathione content in the heart
[9]	Nervous system	40 Hz 7mT for 15min/day for 4 weeks	Brain stroke patients	Oxidative stress can be modulated by EM field.
[10]	Nervous system	50 Hz 1mT	Mice	Enhancement of hippocampal neurogenesis and proliferation of embryonic neural stem cells.

[1] High Voltage Eng. 201642, 2519–2527; [2] Int. J. Rad. Biol. 2018, 94, 909–917; [3] Bioelectromagnetics 2006, 27, 127–131; [4] Sustainability 2018, 10, 2789. [5] Bioelectromagnetics 2000, 21, 608–614; [6] Bioelectromagnetics 2003, 24, 75–8; [7] Arch. Environ. Occup. Health 2012, 67, 65–71. [8] Int. J. Radiat. Biol. 2010, 86, 1088–1094; [9] Bioelectromagnetics 2017, 38, 386–396. [10] PloS One 2016, 11, e0150923.

Thank you again for your valuable comments and suggestions!